# Life Cycle Assessment of Carbon Footprint of Green Tea Produced by Smallholder Farmers in Shaanxi Province of China

Mingbao He [1,2], Yingchun Li [1,*], Shixiang Zong [2,*], Kuo Li [1], Xue Han [1] and Mingyue Zhao [1]

1 Institute of Environment and Sustainable Development in Agriculture, Chinese Academy of Agricultural Sciences, Beijing 100081, China
2 Beijing Key Laboratory for Forest Pest Control, Beijing Forestry University, Beijing 100083, China
* Correspondence: liyingchun@caas.cn (Y.L.); zongshixiang@bjfu.edu.cn (S.Z.); Tel.: +86-10-82105985 (Y.L.)

**Abstract:** China is a major producer of green tea, and most of its green tea production comes from small farmers. Accessing the carbon emission status of this group can provide data support and a decision-making basis for the realization of carbon neutrality in China's tea industry. In this study, the life cycle assessment method was used to analyze the carbon footprint of green tea produced by smallholder farmers in Liugou Village, Hanzhong City, Shaanxi Province. The results showed that the carbon emission intensity of green tea for its entire life cycle was 32.90 kg $CO_2$eq kg$^{-1}$ dry tea, and the carbon emission intensities of its consumption, processing, and cultivation were 14.90, 7.94, and 6.97 kg $CO_2$eq kg$^{-1}$, respectively. In the processing stage, emissions during steaming and drying accounted for 57%. The use of coal, complicated processing procedures, and older equipment were the main reasons for the high emissions in the processing stage. In the cultivation process, emissions mainly came from fertilizer production and its application in the field. The energy consumption of boiling water resulted in high carbon emissions in the consumption stage. This study suggests that building a scientific fertilization system for tea gardens, optimizing processing equipment and energy utilization structure, and cultivating the concept of low-carbon consumption will be the keys to promoting smallholder farmers to reduce carbon emissions. This study further emphasizes that we should focus on carbon emissions caused by the production processes of small farmers.

**Keywords:** green tea; carbon footprint; smallholder farmers; life cycle assessment

## 1. Introduction

Tea (*Camellia sinensis*), the world's most popular non-alcoholic beverage, has vast global market demand. Globally, 69% of smallholder farmers are involved in tea production, and these are mainly located in developing countries [1]. Approximately 60–85% of the tea in China is produced by smallholder farmers [2]. Smallholder farmers usually use their families as production units and rely on that internal labor force for production activities. This method of production is characterized by human capital, assets, small land scale, and a high proportion of agricultural income to total household income [3]. However, as they tend to be limited by their technical and intellectual background, smallholder farmers, in reality, often have the characteristics of low production efficiency and high emissions [4,5], and the hidden indirect costs of household labor are often not considered. In addition, tea, as a labor-intensive industry, significantly influences local economic growth [6]. Tea is widely cultivated in 22 provinces or municipalities in China, and the planting area and yield have steadily increased with the support of local governments [7]. With the continuous expansion of the scale of tea production, the problem of environmental sustainability in tea production should be given more attention. Currently, the issue of carbon emissions has been a gradual concern in various fields. Some studies have shown that consumers in most countries welcome the disclosure of product carbon emission information, while disclosing product carbon footprints also meets some people's needs (such as environmentalists and



high-income people) [8]. In the era of advocating low-carbon production, it is necessary to quantify and analyze the carbon emissions of tea produced by smallholder farmers.

Tea is a perennial leaf plant that requires periodic nitrogen supplementation. Hence, farmers generally practice applying excessive nitrogen fertilizer. However, nitrogen fertilizer is one of the main sources of greenhouse gas emissions in China's tea plantations [9]. Nitrogen oxide ($N_2O$) emissions from nitrogen fertilization in China's tea plantations account for 10% of the total $N_2O$ emissions from the farmland system [10]. At the same time, excessive use of chemical fertilizers may increase market demand for fertilizers and stimulate more emissions from upstream energy consumption. In addition, the nitrogen use efficiency of tea plants in China is only 25–30% [11]. Excessive use of nitrogen fertilizer will lead to increased emissions and harm the planting environment, such as aggravating soil acidification, water eutrophication, and soil nutrient loss [12]. In terms of tea processing production lines, there are still problems in China's tea processing, such as an unreasonable energy structure and low proportion of clean energy, mainly coal and firewood, as the primary fuel for fixation and drying processes. These two processes have enormous energy consumption in the whole processing production line [13–15]. Therefore, exploring the characteristics of carbon emissions in each stage of tea production is the key to understanding the source of carbon emissions for smallholder farmers.

Carbon footprint analysis can quantify the carbon emissions of products and track the emission sources, and this has been applied in many fields. However, from the perspective of previous research, there is no uniform definition of carbon footprint [16,17]. Therefore, we use the definition of a product's carbon footprint given in the ISO 14067: 2018 standard, "sum of GHG emissions and GHG removals in a product system, expressed as carbon-dioxide equivalents ($CO_2eq$) and based on a Life Cycle Assessment (LCA) using the single impact category of climate change" [18]. In LCA studies, LCA methods can be divided into process-based LCA, Economic input-output LCA, or hybrid LCA. For products, the LCA refers to the environmental impact of a product from birth to disposal, which generally includes raw material acquisition, processing, consumption, and treatment [19]. Previous studies have used process-based LCA for estimating the entire life cycle of GHG emissions for tea in China; for example, Liang et al. [20] accounted for the carbon emissions of tea cultivation and processing. Cheng and Liao [21] accounted for the carbon emissions generated by different processing lines of green tea. Xu et al. [22] carried out the full-life cycle carbon footprint accounting for organic tea. He et al. [9] accounted for the full-life cycle carbon footprint for China's provincial-level green tea production. However, there needs to be more attention paid to the tea production activities of smallholder farmers. Internationally, Kenya [23], India [24,25], Sri Lanka [26,27], and Malawi [28] have accounted for the carbon emissions of their tea through LCA, but all had black tea as the accounting object. Therefore, evaluating the carbon footprint of the whole life cycle of green tea among smallholder farmers in China can fill the relevant research gaps.

This study analyzed the carbon footprint composition of the whole life cycle of green tea (from "cradle to grave") at the smallholder level and thoroughly considered the carbon emissions generated from the inputs of production materials during tea cultivation and processing by smallholders, as well as the carbon sequestration function of tea trees. In this study, we aimed: (1) to quantify the carbon emissions of smallholder green tea production in Shaanxi Province, China; and (2) to explore practical and feasible emission reduction measures for small farmers to cope with climate change and help them enhance adaptability. Thus, the findings of this study are expected to provide information for policy-makers and researchers to use to find a decision basis for developing future carbon-neutral pathways in China's tea industry. It also provides suggestions for developing the green tea industry for smallholder farmers.

## 2. Materials and Methods

Shaanxi Province produces much green tea in China and has a long history of its cultivation [29,30]. The green tea grown in Hanzhong is of the best quality and is favored

by the market [31]. The investigation area of this study includes Liugou Village, Mujiaba Town, Nanzheng District, Hanzhong City, Shaanxi Province (Figure 1). The data from the cultivation stage were mainly obtained through local rural cooperatives. The planting area of cooperative tea gardens is 124 ha. The main tea varieties are old varieties: Pingyang Tezao, Cuifeng, and Nanjiang 4. Processing stage data were obtained by visiting local small-scale processing plants. Other stages and relevant emission data were mainly obtained through the literature and thematic reports.

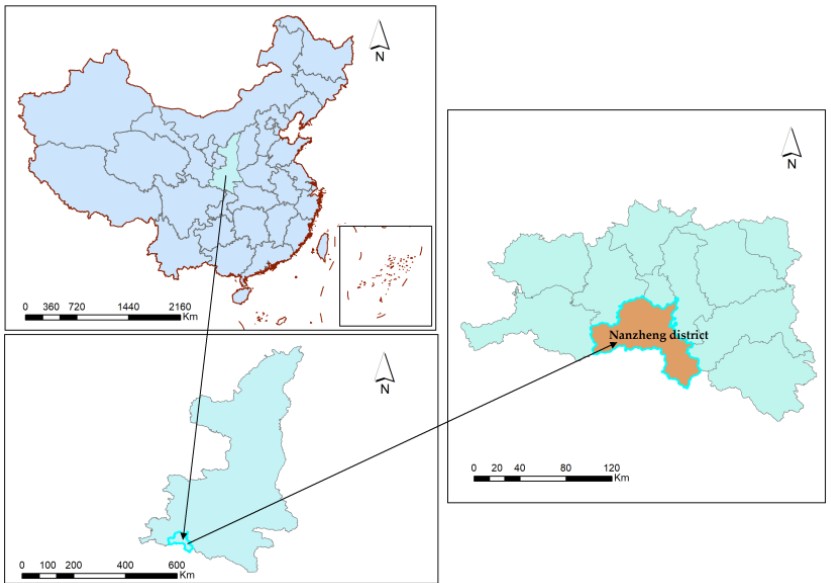

**Figure 1.** Study area.

### 2.1. System Boundary

This study defined the system boundaries and functional units based on ISO14067: 2018 product carbon footprint. The system boundary of carbon footprint accounting in this study is "cradle to grave", the entire product life cycle from cultivation to tea residue disposal (Figure 2). The functional units were the carbon emissions per kg of dry tea (kg $CO_2$eq kg$^{-1}$) and per ha of tea garden (kg $CO_2$eq ha$^{-1}$).

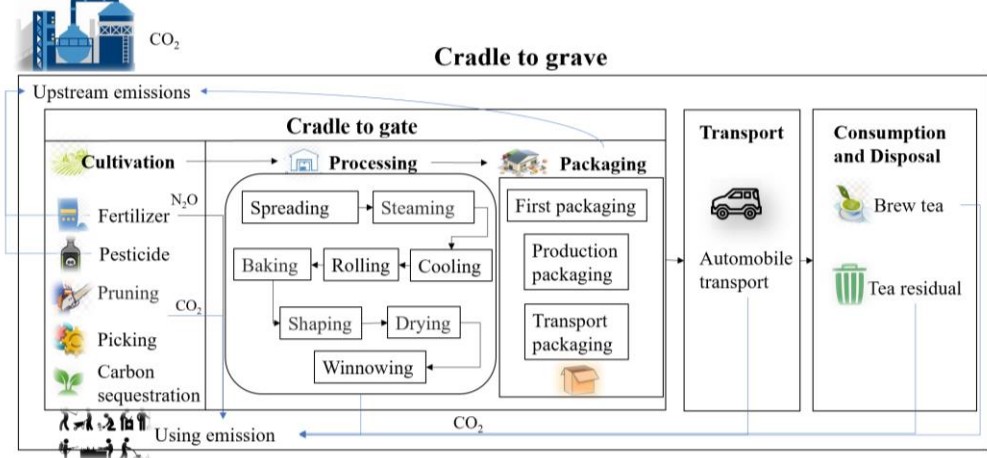

**Figure 2.** LCA system boundary.

The specific considerations for each stage are as follows:

(1)  Cultivation: Emissions from the upstream production of fertilizer and pesticide, as well as emissions from fertilizer application in the field and the use of agricultural machinery in the pruning and harvesting period.

(2)  Processing: Emissions from electricity, coal, and pellets consumed by mechanical equipment in the processing stage.

(3)  Packaging and transportation: Emissions from the production of packaging materials and energy consumption during transportation.

(4)  Consumption and disposal: Emissions from boiling water and tea residue treatment.

*2.2. Data Inventory*

2.2.1. Cultivation

There are two fertilization habits among tea farmers in this study, namely, habit one and habit two. Since the use of pesticides does not directly produce emissions, we calculated the emissions generated by the upstream production of pesticides. Tea picking is performed twice a year, and half of the farmers use machines in the picking process. After harvesting, heavy pruning is performed once a year using a trimmer. Therefore, we considered mechanical equipment as a carbon emission source in pruning and harvesting. Data on inputs and energy consumption at the cultivation stage are shown in Table 1.

**Table 1.** Inputs and energy consumption data at the cultivation stage.

| Stage | Carbon Emissions Source | Activity Data (AD) | Emission Factors (EF) | Reference |
|---|---|---|---|---|
| Cultivation | Practices one (upstream) — Diammonium phosphate | 517.5 kg $P_2O_5$ ha$^{-1}$ | 4.07 kg $CO_2$eq (kg $P_2O_5$)$^{-1}$ | [32] |
| | Urea | 517.5 kg N ha$^{-1}$ | 7.48 kg $CO_2$eq (kg N)$^{-1}$ | |
| | Practices two (upstream) — Ammonium carbonate | 510 kg N ha$^{-1}$ | 7.07 kg $CO_2$eq (kg N)$^{-1}$ | |
| | Phosphate fertilizer | 690 kg $P_2O_5$ ha$^{-1}$ | 2.33 kg $CO_2$eq (kg $P_2O_5$)$^{-1}$ | |
| | Urea | 690 kg N ha$^{-1}$ | 7.48 kg $CO_2$eq (kg N)$^{-1}$ | |
| | Nitrogen fertilizer application — Direct emissions of $N_2O$ | 517.5 kg N ha$^{-1}$ (Practices one); 1200 kg N ha$^{-1}$ (Practices two) | 1.92% | [33] |
| | Nitrogen deposition | | 1% | [34] |
| | Leaching runoff | | 1% | |
| | Chemical pesticide (upstream) — Pesticide (Cypermethrin, 100%) | Twice a year | 54.27 kg $CO_2$eq ha$^{-1}$ | [35] |
| | Germicide (Carbendazim, 60%) | Twice a year | 57.57 kg $CO_2$eq ha$^{-1}$ | |
| | Machinery — Pruning (90%) Picking (50%) | 37.5 L ha$^{-1}$ (once) 30 L ha$^{-1}$ (twice) | 2.925 kg $CO_2$eq kWh$^{-1}$ | [36] |

Note: % in parentheses indicates the proportion of the area used. The fuel emission factors in this article were derived from reference [36], referred to as the guidelines.

2.2.2. Processing

Tea processing has several procedures, including spreading, steaming, cooling, rolling, baking, shaping, drying, and winnowing. Generally, 4.5 kg of fresh leaves produces 1 kg dry tea on average. We collected the energy consumption and the amount of fresh leaf processing per machine equipment in the production line. Information on the energy input of each procedure and emission factors of different energies are shown in Tables 2 and 3.

**Table 2.** Machining phase data sheet.

| Processing | Energy Input and Tea-Processing Capacity of Each Procedure | | | |
|---|---|---|---|---|
| | Electricity (kwh) | Coal (kg h$^{-1}$) | Pellets (kg h$^{-1}$) | Tea-Processing Capacity (kg h$^{-1}$) |
| 1. Spreading | 0.175 | / | / | 75–100 |
| 2. Steaming | 24 | 83.33–100 | 0.83 | 300–400 |
| 3. Cooling | 2.5 | / | / | 400–600 |
| 4. Rolling | 30 | / | / | 145 |
| 5. Baking | 13 | 66.67–83.33 | 75 | 600 |
| 6. Shaping | 5 | / | 15 | 75 |
| 7. Drying | 4.5 | 41.67 | / | 150 |
| 8. Winnowing | 2.5 | / | / | 75 |

**Table 3.** Emission factors of different energy sources.

| Energy | EF | Unit | Reference |
|---|---|---|---|
| Electricity | 0.667 | kg CO$_2$eq kWh$^{-1}$ | [37] |
| Coal | 1.74 | kg CO$_2$eq kg$^{-1}$ | [36,38] |
| Pellets | 0.968 | kg CO$_2$eq kg$^{-1}$ | |

### 2.2.3. Packaging and Transport

The packaging stage could be categorized into primary packaging, transportation packaging, and product packaging (Table 4). Primary packaging refers to the plastic woven bags used in tea factories to hold semi-made tea. Transportation packaging refers to the corrugated boxes in which tea is transported to sale places. Product packaging refers to the packaging for tea sales, including boxes, bags, cans, etc. This study considered the average emission factors of product packaging (Table 4).

**Table 4.** Packaging phase data sheet.

| Packaging Category | AD | EF | Reference |
|---|---|---|---|
| Primary packaging | 45 kg tea per bag | 0.37 kg CO$_2$eq per bag | [39] |
| Transportation packaging | 25 kg tea per box | 1.90 kg CO$_2$eq per box | [40] |
| Product packaging | 1 kg per tea | 2.30 kg CO$_2$eq kg$^{-1}$ per kg tea | [22] |

Note: The specifications of corrugated cartons for transport packaging are 60 × 43 × 32, weighing approximately 1 kg, and its emission factor can be calculated according to the research of Han [40].

Carbon emissions in the entire transportation stage are mainly determined by distance and the means of transport. The actual transport distance was divided into two parts: the distance from the tea plant to the factory and the distance from the factory to the sale place. However, the tea garden was near to the processing factory, and the emissions generated by this portion of the journey were not considered in this study. We only calculated the emissions in the transport stage from the processing plant to the sale place, and the means of transportation were cars. The details are shown in Table 5.

**Table 5.** Transport phase data sheet.

| Destination (Province) | Bazhong (Sichuan) | Hanzhong (Shaanxi) | Ankang (Shaanxi) | Xi'an (Shaanxi) | Longnan (Gansu) |
|---|---|---|---|---|---|
| Freight volume | 40% | 20% | 10% | 20% | 10% |
| Haul distance | 156.10 km | 28.30 km | 241.70 km | 312.20 km | 315.50 km |
| EF | 0.29 kg CO$_2$eq km$^{-1}$ [41] | | | | |

Note: It was learned from farmers that the average cargo volume per shipment was 450 kg, and the means of transport was the Baojun 310 fuel vehicle.

2.2.4. Consumption and Disposal

In China, electric kettles are mainly used to boil water, and thermal power generation is the primary power supply method. This study focused on the carbon emissions generated by electricity consumption. The personal practices of consumers during the tea-brewing process determine the amount of tea and hot water used, which will affect the emission results. In this study, we made reasonable assumptions based on research, which showed that average tea consumption in China is 3 to 5 g per person, and the amount of water used is approximately 4.54 cups [42]. Boiling water generally comes from household electric kettles, so the electricity consumption of kettles was obtained from the measured results in relevant literature [43]. Therefore, we assumed the consumer's tea consumption to be 4 g per instance, and their water use to be 4 cups. That is, the amount of hot water per instance was assumed to be 800 mL. The consumption places were the sale destinations (Shaanxi, Sichuan, and Gansu). They are located in the northwestern and central regions of China, according to the power grid area (Table 6). The consumption weights of different consumption places were allocated according to the proportion of freight volume (3:2), as shown in Table 5, and the final consumption stage emissions were a weighted average.

**Table 6.** Emission inventory of consumption and disposal.

| Category | Value | Unit |
|---|---|---|
| Consumption per instance | 4 | g |
| Amount of hot water | 800 | mL |
| Electricity consumption | 0.122 | Kwh L$^{-1}$ |
| **Emission sources** | **Value** | **Unit** |
| Northwest Regional Power Grid (Shaanxi, Gansu) | 0.667 | kg $CO_2$eq kWh$^{-1}$ |
| Central China Regional Power Grid (Sichuan) | 0.526 | kg $CO_2$eq kWh$^{-1}$ |
| Landfill | 0.674 | kg $CO_2$eq kg$^{-1}$ |
| Incineration | 0.229 | kg $CO_2$eq kg$^{-1}$ |

In this study, waste treatment refers to the treatment of waste tea residue, which is divided into incineration treatment and landfill treatment. According to the "China Statistical Yearbook" [7], the treatment volume of the two treatment methods in the aforementioned consumption areas accounted for an average of 50% each. The emission factors for both types of treatment were derived from Bian et al. [44], and we calculated the tea residue according to the classification of food.

*2.3. Carbon Footprint Calculation Method*

The calculation method of field $N_2O$ emissions was derived from the 2006 IPCC Guidelines for National Greenhouse Gas Inventories [45], and the emissions of energy use were calculated based on the NRDC Guidelines [36].

2.3.1. $CO_2$ Emissions

Emissions from the use of materials and their upstream production in the cultivation, processing, packaging, transportation, consumption, and treatment of tea, as well as $CO_2$ emissions from the combustion of electricity or fossil fuels, were calculated using the following formula:

$$E_i = \sum_i AD_{ij} \times EF_{ij} \tag{1}$$

where $E_i$ is the input total carbon emissions (kg $CO_2$eq ha$^{-1}$); $AD_{ij}$ is the activity level data, that is, the consumption of materials or energy (kg or kWh, respectively); $EF_{ij}$ is the emission factor (carbon emissions per unit); and $i$ and $j$ represent a certain stage and the input of materials or energy, respectively.

2.3.2. N$_2$O Emissions

Nitrogen fertilizer application at the cultivation stage produces N$_2$O emissions, which can be divided into direct and indirect emissions. Direct emissions refer to the N$_2$O release from soil caused by synthetic fertilizer. Indirect N$_2$O emissions are caused by atmospheric nitrogen deposition, nitrogen leaching, and runoff losses.

$$E_d = AD \times EF_d \times \frac{44}{28} \times 265 \tag{2}$$

where $E_d$ represents the direct N$_2$O emissions of tea gardens every year, $AD$ represents the nitrogen fertilizer nutrient consumption (kg N kg$^{-1}$), $EF_d$ is the direct N$_2$O emission factor kg CO$_2$eq ha$^{-1}$, 44/28 is the ratio of the molecular weight of nitrous oxide to nitrogen, and 265 is the 100-year warming potential value of N$_2$O [46].

Indirect emission formulas:

$$E_{ATD} = AD \times 11\% \times EF_{\text{deposition}} \times \frac{44}{28} \times 265 \tag{3}$$

$$E_L = AD \times 24\% \times EF_{leaching} \times \frac{44}{28} \times 265 \tag{4}$$

$$E_{N_2O} = E_d + E_{ATD} + E_L \tag{5}$$

where $E_{ATD}$ represents the N$_2$O emissions caused by atmospheric sedimentation (kg CO$_2$eq ha$^{-1}$), 11% is the proportion of nitrogen fertilizer volatilization, and $EF_{\text{deposition}}$ is the emission factor for atmospheric deposition. $E_L$ represents the N$_2$O emissions generated by leaching runoff (kg CO$_2$eq ha$^{-1}$), 24% is the proportion of nitrogen fertilizer loss, and $EF_{leaching}$ is the emission factor for N leaching and runoff. $E_{N_2O}$ represents the field's total N$_2$O emissions after nitrogen fertilizer application (kg CO$_2$eq ha$^{-1}$). The ratio of nitrogen fertilizer volatilization and loss is from the IPCC 2019 [34]. The 100-year warming potential value of N$_2$O is 265 [46].

2.3.3. Carbon Storage of Tea Plants

As a perennial woody plant, tea plants have certain carbon sequestration potential, which can be divided into two parts: the CO$_2$ absorbed during photosynthesis in the tea plants and the fixation of soil organic carbon. The carbon dioxide fixed by tea plants can be calculated by biomass, which is key to assessing carbon sinks in artificial tea plantation ecosystems [47]. Since the economic coefficient of tea plants is only 0.2 and the pruned branches and leaves enter the tea garden ecosystem by returning to the field, we can ignore the biomass lost by tea plants due to picking and pruning. Tea biomass was calculated according to the tea plant growth model for biomass, expressed as Equation (6) [47], and this model was considered to be the general growth model of most tea gardens [48]. The results included above-ground and underground parts, calculated based on an average root-to-crown ratio of 0.5 (Equation (7)). The soil carbon sequestration in the tea gardens was not considered in this study, as the soil carbon storage rate of the tea gardens gradually weakens with the tea's age until it approaches zero [49]. The final calculated biomass was converted into CO$_2$ through Equation (8), which indicated the average annual CO$_2$ absorption per unit area of tea plants.

$$M_{\text{Plant}} = -14.95 + 56.3 \left( 1 - e^{-0.27t} \right) \tag{6}$$

$$M_{\text{tall}} = M_{\text{Plant}} \times (1 + 0.5) \tag{7}$$

$$C_{\text{Plant}} = \frac{M_{\text{tall}} \times 0.5 \times \frac{44}{12}}{30} \tag{8}$$

where $M_{\text{Plant}}$ is the above-ground biomass (kg ha$^{-1}$); t is the age of the tea plant, assumed to be 35 years in this study; $M_{\text{tall}}$ is the total biocarbon sequestration of the tea plant (kg ha$^{-1}$),

above and below ground; $C_{Plant}$ is the average annual $CO_2$ absorbed by tea plants during harvesting (kg $CO_2$eq ha$^{-1}$); 0.5 is the carbon content of tea biomass; and 44/12 is the $CO_2$ conversion coefficient. Because artificially planted tea plants are generally picked from the 5th year, the total period of picking generally lasts 30 years.

### 2.3.4. Carbon Emissions of Tea per Unit

Equations (9)–(11)

$$E_A = E_{i,a} + E_{N_2O} - C_{Plant} \tag{9}$$

$$E_{agr} = \frac{E_A}{Y} \tag{10}$$

$$E = E_{agr} + E_{pro} + E_{pac} + E_{tra} + E_{con} + E_{dis} \tag{11}$$

where $E_A$ represents the carbon footprint of the tea garden (kg $CO_2$eq ha$^{-1}$); $E_{i,a}$ represents the carbon emission of input in cultivation stage (kg $CO_2$eq ha$^{-1}$); $E_{agr}$ represents the carbon emissions per kg dry tea in the cultivation stage (kg $CO_2$eq kg$^{-1}$); $Y$ represents the yield (kg ha$^{-1}$); $E$ represents the whole carbon footprint of the tea (kg $CO_2$eq kg$^{-1}$); and $E_{pro}$, $E_{pac}$, $E_{tra}$, $E_{con}$, and $E_{dis}$ represent the carbon emissions per kg of tea during processing, packaging, transport, consumption, and disposal, respectively.

## 3. Results

### *3.1. Carbon Emissions from Each Stage*

Regarding "cradle to grave", tea cultivation, processing, and consumption were the primary sources of carbon emissions (Figure 3). The life cycle carbon footprint of green tea was 32.90 kg $CO_2$eq kg$^{-1}$ dry tea, of which the consumption stage contributed 14.90 kg $CO_2$eq kg$^{-1}$, accounting for 45% of the total emissions. The second and third highest carbon emissions were from processing and cultivation, with 7.94 and 6.97 kg $CO_2$eq kg$^{-1}$, accounting for 24% and 21% of the total emissions, respectively. The lowest carbon footprints were in the transportation and disposal stages, which accounted for only 3% of the total emissions. The remaining 7% of emissions came from the packaging stage.

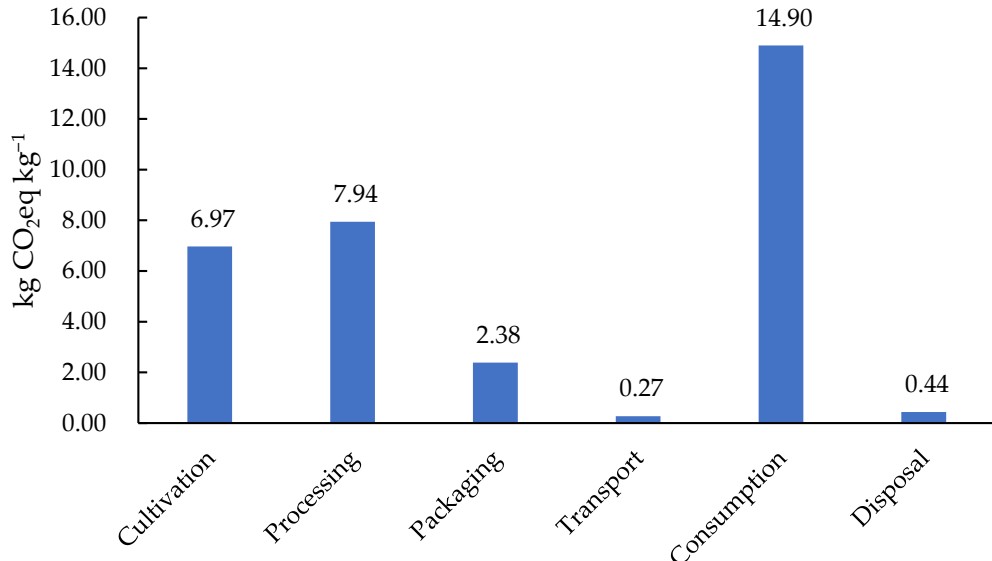

**Figure 3.** Carbon emissions from each stage.

### *3.2. Carbon Footprint of the Cultivation Stage*

In the cultivation stage, approximately 49% of greenhouse gas emissions came from nitrogen fertilizers application, and 48% of the carbon emissions were generated at the upstream production of fertilizers. Different fertilization habits directly affect carbon emissions. Habit two produced 39% higher emissions than habit one (Table 7). The carbon

emissions from machinery and pesticide use were relatively small. Overall, the average carbon emissions per hectare of tea gardens were 13.37 t $CO_2$eq. The average carbon sequestration of tea plants per hectare was 3.79 t $CO_2$eq, and the net emissions were 9.58 t $CO_2$eq ha$^{-1}$ (Figure 4).

**Table 7.** Composition of carbon emissions during the cultivation stage (kg $CO_2$eq ha$^{-1}$).

| Input | Carbon Emission Source | Emissions | Total |
|---|---|---|---|
| Fertilizer | Practice one (upstream) | 5977.13 | 10,920.74 |
| | $N_2O$ emissions | 9887.22 | |
| | Practice two (upstream) | 4943.61 | 15,159.47 |
| | $N_2O$ emissions | 8167.71 | |
| | Upstream (average) | 6484.45 | 13,040.10 |
| | $N_2O$ emissions (average) | 6555.66 | |
| Chemical pesticide | Cypermethrin | 108.53 | 174.35 |
| | Carbendazim | 65.81 | |
| Machinery | Pruning (90%) | 74.04 | 156.30 |
| | Picking (50%) | 82.27 | |

Note: The % in parentheses is the area applied.

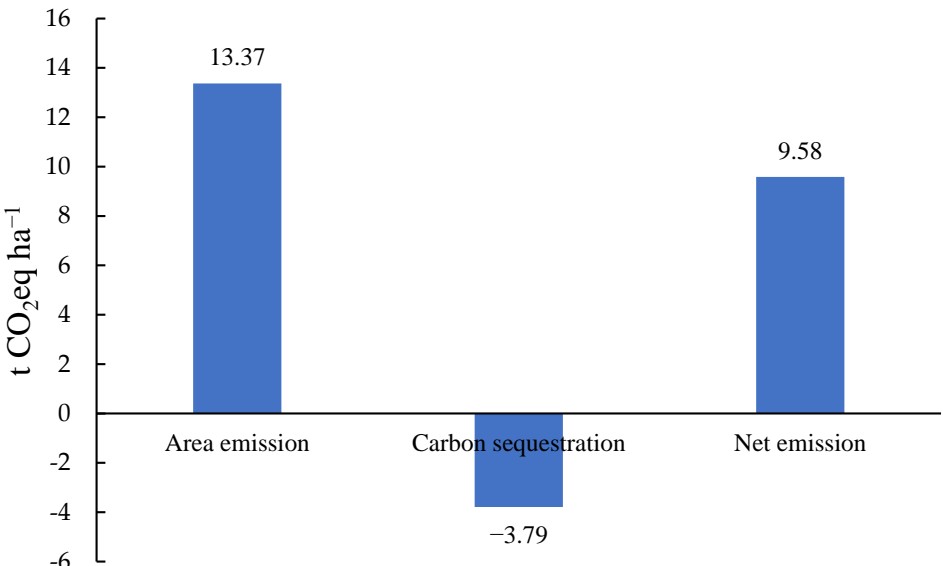

**Figure 4.** Carbon emissions per hectare of tea garden.

### 3.3. Carbon Footprint of the Processing Stage

The carbon footprint and emissions from the energy consumption of each step in the processing stage are shown in Figure 4. Among all processing steps, the carbon emissions per unit in the steaming and drying process were the highest, both 2.27 kg $CO_2$eq kg$^{-1}$, followed by the baking process with 1.59 kg $CO_2$eq kg$^{-1}$. The spreading process had the lowest emissions. In terms of the proportion of emissions from energy consumption, more than 90% of steaming and drying processing emissions were from coal use (Figure 5). Emissions from pellets ranged between coal and electricity. In addition, electrical energy was applied in all steps, and the emissions generated by electricity consumption were also at a minimum. The lowest emissions were mainly based on the use of electric energy.

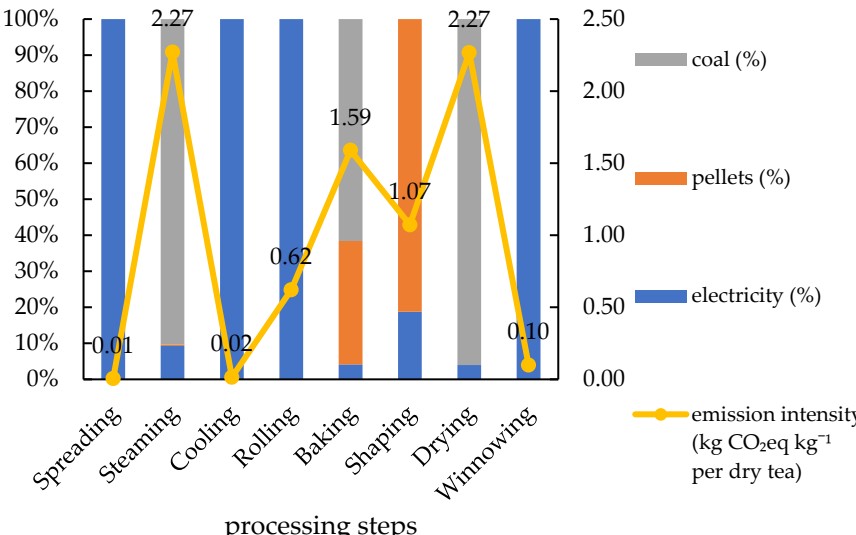

**Figure 5.** Carbon footprint of the processing stage and proportions of each energy source used at different steps.

## 4. Discussion

### 4.1. Carbon Footprint of the Production System

This study used field survey data to estimate the CF of green tea produced by small-holders in Shaanxi Province (Figure 1). Production systems, from cultivation to packaging, accounted for 45% of total emissions throughout the life cycle. Due to the substantial amount of coal use and complicated processing steps, emissions from the processing stage were higher than those from cultivation (Figure 3). Among the two most carbon-emitting processing steps, the use of coal contributed to the high emissions (Figure 4). In addition, compared with simple green tea, two fixations increased the fuel input and, thus, the emission intensity.

A study reported that 1 kg of green tea emitted 3.45 kg $CO_2$eq based on an assumption of calorific value, which was simply converted into energy consumption [50]. However, that study ignored heat loss in the intermediate process of green tea processing, which may lead to low estimates. Cheng and Liao [21] concluded that the carbon emissions per kg of green tea were 4.53 kg $CO_2$eq. Nevertheless, this class of green tea only involved several steps, including fixation, shaping, and drying; therefore, the study results were lower than those of our study. In addition, Xu et al. [22] estimated the carbon footprint values of famous green tea (Longjing and Wuyang Chunyu) in the processing stage, with values of 9.7 and 6.3 kg $CO_2$eq kg$^{-1}$, respectively, close to the values of our study. However, there were significant differences in energy structure and production scale from this study. From the perspective of the energy structure of the processing stage, the energy structure of Xu et al.'s research was mainly electricity, and the emissions were far lower than those of coal. The analysis showed that green tea produced by ordinary small workshops might produce higher emissions than enterprise-level production.

The processing stage of green tea implied the input of machinery and equipment, indicating that the processing production of farmers had achieved a high mechanized level, reduced labor, and increased production capacity. However, our on-site visit found that most production types of equipment were old-style and frequently encountered mechanical failures. In addition, the lack of continuity between devices, mainly the single-machine heating operation of "one machine and one stove," caused the waste of intermediate process energy, resulting in more carbon emissions. Furthermore, the lag of the energy structure of the processing stage was also one of the main reasons for high emissions. There is still a significant dependence on the use of coal, which is mainly related to processing equipment. Chen et al. [51] set up a waste heat recovery scheme that can reduce energy demand by 21% for tea processing while reducing input and greenhouse gas emissions. Therefore,

improving the efficiency of heat energy utilization has a greater role in promoting the emission reduction of the processing stage. However, improving the efficiency of heat energy utilization may mean that there is a need of processing equipment upgrades or other inputs, which is a high economic burden for smallholder farmers and will affect the enthusiasm of farmers. It is recommended that the government introduce subsidy policies to encourage smallholder farmers to replace old equipment and provide technical guidance to optimize processing production lines through training and other ways to help farmers establish energy conservation and emission reduction awareness.

The cultivation stage is another primary source of emissions in the production system of green tea. Compared with other studies, the carbon emissions of tea gardens had a higher emission level than other agricultural products. The calculation results of Chen et al. [52] showed that the average carbon emissions of rice, cotton, and tobacco in China were 9.5, 4.4, and 2.0 t $CO_2$eq ha$^{-1}$, respectively, which were lower than the results of this study (9.58 t $CO_2$eq ha$^{-1}$). The largest source of carbon emissions of all inputs in the cultivation stage mainly came from chemical fertilizers. Nitrogen fertilizer generated the greatest carbon emissions among the chemical fertilizers, accounting for 89% of the total emissions from cultivation (including upstream and application emissions). The average input of local nitrogen fertilizer was 686.25 kg N·ha$^{-1}$·a$^{-1}$, which was much higher than the recommended fertilization amount of 150-300 kg N·ha$^{-1}$ a$^{-1}$ [53] and the national average of 491 kg N·ha$^{-1}$·a$^{-1}$ [33]. Excessive use of nitrogen fertilizer will adversely affect the growing environment of tea plants, such as soil acidification and $N_2O$ emissions. Although tea plants can grow in acidic soils, the growth of tea plants and the quality of tea leaves are affected when the soil pH < 4.5 [53]. Significant emissions of $N_2O$ gases exacerbate global warming and increase the probability of more extreme weather events [54]. As a perennial woody plant, climate change will affect the yield and quality of tea and weaken its adaptability to climate change [55,56]. Therefore, experts have focused on reducing the amount of chemical fertilizer and increasing the input of organic fertilizer. Studies have observed that using rapeseed organic fertilizer in tea gardens can significantly reduce the runoff loss of nitrogen compared with compound fertilizer and increase soil organic matter [57]. In recent years, novel fertilizers have been widely promoted and applied. Field experiments have shown that using novel fertilizers for major food crops (wheat, corn, rice, etc.) can improve crop yield and nitrogen use efficiency, and reduce economic inputs and nitrogen losses [58]. Smallholder farmers have switched from traditional to organic farming, which can reduce the carbon emissions of the cultivation stage while also increasing economic benefits [5]. However, some studies have suggested that the transition to organic farming will reduce tea production and increase labor inputs, which will not be enough to fully meet farmer livelihoods [59]. Therefore, we should carefully compare the pros and cons before choosing emission reduction measures, promoting and applying new low-carbon fertilizers, or developing a new, low-carbon green tea development model.

*4.2. Carbon Footprint of the Consumer System*

The consumption stage generated the highest carbon emissions in the life cycle of green tea products, mainly caused by the much higher proportion of hot water. In this study, brewing 4 g of green tea requires a total of 800 mL of hot water, meaning drinking 1 kg of green tea requires 200 L of hot water. Therefore, in the consumption stage, attention should be given to improving the energy efficiency and structure of boiling water. Studies have found that controlling water temperature and improving the efficiency of boiling kettles can reduce carbon emissions, in which improving kettle efficiency was the most effective way to reduce emissions [43]. In addition, consumers' high demands for tea quality will have a particular impact on the emissions of tea production. Some studies have indicated that the preference of domestic consumers for high-quality tea is one of the main reasons for the high emissions of tea in China. High-quality requirements reduce the production of tea, such as only tender leaves or buds being picked [60]. From a comparison of different studies, the emissions of famous green tea were higher than those of ordinary green tea

(Figure 6). Supposing that consumers have increasing requirements for higher-quality green tea products. In this case, it could increase the emission reduction pressure at the production system. Therefore, consumers are encouraged to drink low-carbon labeled tea.

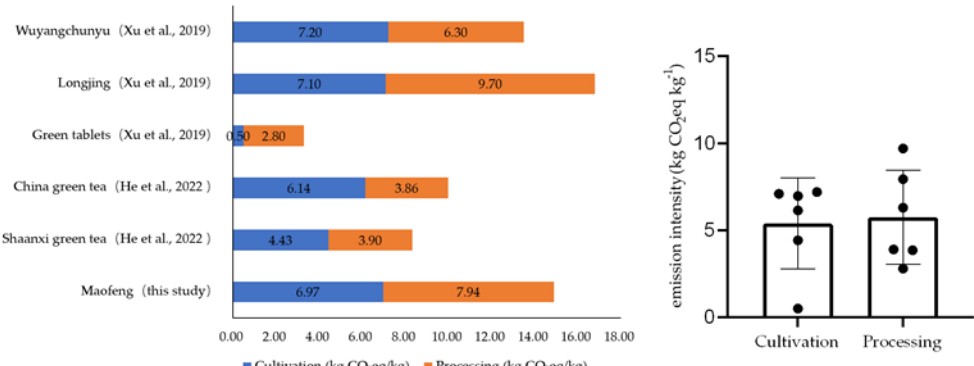

**Figure 6.** Comparison of carbon emissions from cultivation and processing stage in different studies. Note: In addition to the green tablets in the study by Xu et al. [22], the rest were famous green tea. The green tea studied by He et al. [9] was ordinary green tea. This article examined famous green tea; the right figure values are means ± sd.

*4.3. Limitations of This Study*

This study conducted a detailed study of the carbon footprint of green tea products, but there are still some limitations in the research methods. First of all, agricultural production activities are highly dependent on natural resources and have strong spatial heterogeneity. Similarly, production activities on environmental pollution will also have strong regional characteristics [61]. However, we ignore the study and discussion of regional spatial heterogeneity, which can further study in future research. Secondly, on the system boundary, limited by the availability of relevant data, we ignore other possible emission sources, such as emissions from seed production and processing. Finally, we mainly studied the carbon footprint of tea, while other types of environmental impacts, such as ecotoxicity potential (USEtox model) [62] and eutrophication potential (EUTREND model), were not considered [63]. In future studies, these environmental impacts can be incorporated into tea research to more comprehensively evaluate the environmental impact of tea produced by smallholder farmers.

**5. Conclusions**

The carbon footprint of tea produced by smallholder farmers in Liugou Village, Hanzhong City, Shaanxi Province, was calculated for the whole life cycle. The life cycle carbon emissions were 32.90 kg $CO_2$eq kg$^{-1}$ dry tea. The consumption stage was the most prominent emissions source, accounting for 45% of the total emissions. Cultivation, processing, packaging, transportation, and disposal accounted for 21%, 24%, 7%, 1%, and 2%, respectively. The processing stage was the second source of emissions, followed by cultivation, with 7.94 and 6.97 kg $CO_2$eq kg$^{-1}$, respectively. High levels of mechanization, the complexity of processing, old equipment, and the use of a large amount of coal were the main reasons for the high emissions in the processing stage. Steaming and drying were the largest sources of emissions in the processing stage, accounting for 57% of the carbon emissions of the entire processing stage. $N_2O$ emissions from nitrogen fertilizer use contributed 49% of carbon emissions from the cultivation stage. The upstream end of fertilizer products contributed 48%, while pesticides and agricultural machinery investment accounted for only a tiny proportion (approximately 1%). In summary, this study found that the main way to reduce emissions in the production system of green tea products is to reduce the use of chemical fertilizers and optimize processing to improve the efficiency of heat energy utilization. Improving consumer concepts of low-carbon consumption is critical to reducing emissions in the consumer system. In the future, climate change may

have more uncertain impacts on the tea production of smallholder farmers. Society should help tea producers establish awareness of energy conservation and emission reduction. More research on the sustainable development path of green tea for smallholder farmers will be helpful.

**Author Contributions:** Conceptualization, Y.L. and S.Z.; methodology and software, K.L.; validation and formal analysis, M.Z.; investigation and data curation, M.H. and X.H.; writing—original draft preparation, M.H.; writing—review and editing, Y.L.; visualization and supervision, Y.L. and S.Z. All authors have read and agreed to the published version of the manuscript.

**Funding:** This research was funded by the National Key R&D Program of China (2019YFA0607403), the National Natural Science Foundation of China (D41105115), Chinese Academy of Agricultural Sciences Central Public-interest Scientific Institution Basal Research Fund (BSRF202202) and the Agricultural Science and Technology Innovation Program (ASTIP).

**Institutional Review Board Statement:** Not applicable.

**Informed Consent Statement:** Not applicable.

**Data Availability Statement:** Not applicable.

**Conflicts of Interest:** The authors declare no conflict of interest.

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
