# Peer review of "Life Cycle Assessment of Carbon Footprint of Green Tea Produced by Smallholder Farmers in Shaanxi Province of China"

_agronomy, doi:10.3390/agronomy13020364_

Round 1
Reviewer 1 Report
General: This paper is nicely drafted and easy to read.
Abstract:
Please clearly mention hypothesis, objective, and end abstract with future implications of this study.
Introduction:
Line 26: Scientific names are always in italics; Does water comes under beverage category?
Line 27: reference
Line 34: Often hidden indirect cost of family labor is not accounted
Line 47: requires periodic nitrogen supplementation
Line 53: Better use term, “ Nitrogen use efficiency”
Line 69-73: It would be great if authors can highlight different models of LCA used worldwide in C footprint analysis. (Process based model or hybrid model or social inventory based-IO model, USEtox (Hauschild et al., 2008) for toxicity, the IPCC model (IPCC, 2007) for climate change and EUTREND (Van Jaarsveld, 1995) to classify fate, exposure and effect at each step of the process. If not then may be in the discussion section, which is more robust for tea plantation.
Line 82: Clearly highlight hypothesis, objectives, and end with future implications of this work after objectives.
M&M:
Line 88-90: Map with growing area will help readers
Figure 1: Improve figure resolution
Line 117: Please replace habit word to practices in the entire manuscript
Line 247: Any statistical comparison between two practices of fertilization?
Results:
Figure fonts should match with manuscript (Palatino Linotype)
Figure 3: Instead of red C sequestration can be shown as -3.79
Discussion:
I would suggest the authors to have a more supported discussion with references considering the main point: The limitations of method and considerations when to apply the studied methodology and then the potential next steps or further investigation to address these limitations.
References: Please double check the style of references and missing one
Reviewer 2 Report
The research presented in this manuscript is interesting and provides knowledge for mitigating climate change. In the light of the results of the studies, it should be recommended that the government introduce a subsidy policy to encourage smallholder farmers to replace old equipment and optimize tea processing lines. Farmers need to realize of the importance of saving energy and reducing emissions.
The manuscript meets the requirements of Agronomy and is suitable for publication, only a few inaccuracies need to be corrected. I have pointed them out in the manuscript.
Important: the data on the right-hand side of Figure 5 have been statistically evaluated. The method of statistical analysis used should be indicated.
